# Fast Classification Model Based on Genetic Algorithm and XGBoost-RandomForest Stacking Model

1st Yanliang Zhou
*College of Marine Electrical Engineering*
*Dalian Maritime University*
Dalian, China
yanliangzhou@dlmu.edu.cn

2nd Tianhe Liu
*College of Marine Electrical Engineering*
*Dalian Maritime University*
Dalian, China
tianheliu@dlmu.edu.cn

3rd Jiawen Wang
*College of Marine Electrical Engineering*
*Dalian Maritime University*
Dalian, China
wjw18963483217@126.com

4th Jie Cheng
*College of Marine Electrical Engineering*
*Dalian Maritime University*
Dalian, China
15958505778@163.com

*Abstract*—The balance between calculation accuracy and running time is a problem in effectively utilizing the stacking model. In this paper, a hybrid ensemble learning model is introduced to improve computational accuracy, and genetic algorithm is used to select features in the training process to reduce running time. The model is built by testing a single model, selecting a model with good performance to form a stack model, and then using genetic algorithm to select features on the stacking model. The proposed model demonstrates superior comprehensive performance compared to both individual models and stacking models, as verified by analysis of three datasets.

*Index Terms*—Genetic algorithm; Random forest; XGBoost; Stacking model

## I. INTRODUCTION

In machine learning, a stacking model is a very flexible model that can be combined in different ways. It takes prediction results of one or more base models as new features, and uses those for new model training. This newly trained model, also known as meta-model, is used to make predictions [1].

Wolpert first proposes the theory of stacking models in 1992 [2], and in 1996, Leo Breiman improves the original stacking model, leading to the modern stacking model that now uses internal k-fold cross-validation [3]. To this day, stacking models plays an essential role in different fields. Tao Peng uses support vector machines(SVM), XGBoost(XG) and Light Gradient Boosting Machine for stacking and uses the models for fault diagnosis of electric motors [4]; Xiaofeng Dong uses the AdaBoost-RandomForest stacking model for predicting likelihood of diabetic readmission [5]; Yao Jinwei uses an optimized three-layer stacking model to estimate traffic flow [6]. However, there are still many shortcomings of the stacking model that still need to be addressed, among which complexity is the most crucial issue for stacking model. The computational load and complexity may rise with the growth of the number of layers and models in a stacking model. Existing methods to reduce running time, such as parameter tuning and data sampling, may change the generalization performance of the model. Therefore, in practical applications, it is necessary to choose a suitable model and select a specific optimization method to balance running time and performance of the superimposed model. Thus ensuring that the model can be trained within resources and time frame allowed.

In order to reduce running time of stacking models, feature selection methods can be used for optimization. Among them, the genetic algorithm (GA) is one of the most commonly used optimization algorithms for feature selection. The theory of using GA for feature selection is proposed by Haleh Vafaie and Kenneth De Jong in 1992, in which the authors proposed that combining the genetic algorithm for feature selection with the desired fitness function could reduce the number of features while improving accuracy of the model [7].

Motivated by the discussion above, this paper primarily proposes a GA-XGBoost-RandomForest stacking model, which is optimized using GA and utilizes XG and Random forest (RF) for stacking. This approach leverages computational efficiency of XG to minimize running time. As a model with strong generalization ability, RF performs well on different datasets and can complete the final output task well. GA performs feature selection in the basic model training of stacking models to reduce overall running time.

The main contribution of this paper are as follows:

- XG-RF stacking model is proposed to solve the problem of long running time while improving accuracy of stacking model. The model can maintain a certain precision without increasing the number of layers and the number of models per layer.
- GA is used for feature selection to optimize the training process of XG, thereby reducing the running time of stacking models and improving the computational accuracy of the model.

The rest of this paper is organized as follows. Section 2 introduces the main algorithms used in this paper. Section 3 introduces the structure and operation flow of the model. Experiments are conducted and the experimental results are discussed in section 4. Section 5 concludes this paper.

## II. RELATED WORK

### A. Stacking method

Stacking is an effective ensemble method, and stacking models generally consist of two layers [8]. Predictions generated by the first layer using various machine learning algorithms are used as inputs to the second layer learning algorithm. This second layer algorithm is trained to optimize combined model predictions to form a new set of predictions [9].

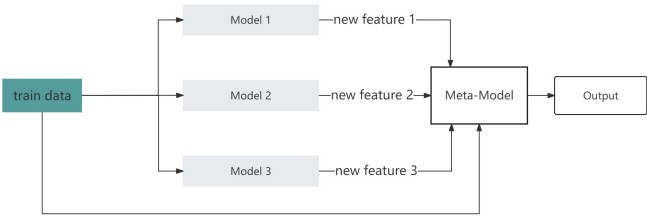

Fig. 1. Stacking Method Flowchart

In the stacking model, the basic model of first layer can include a variety of different types of machine learning algorithms, such as decision trees, SVM, and KNN, there are different characteristics and performance in each of them. By combining them, stacking models can take advantage of their combined strengths to improve performance of the overall model. A meta-model is usually a simple model that combines predictions of first layer model and generates final output [10]. This layered stacking method can reduce overfitting to a certain extent, improve the generalization ability of the model, and make the stacking model perform well when dealing with complex tasks [11].

### B. Genetic algorithms

GA is a heuristic algorithm that borrows its principles from Darwinian evolutionary ideas [12]. The idea is to optimize initial population through multiple iterations and end up with the most adapted individuals [13]. GA first randomly generates an initial population and evaluates each individual in population using a pre-set fitness function to measure how well it matches ideal solution. Standard methods uses in the iterative process include mutation, intersection, and reproduction. The whole process continues for multiple generations until individual's fitness reaches a target value or a given number of generations are reached [14].

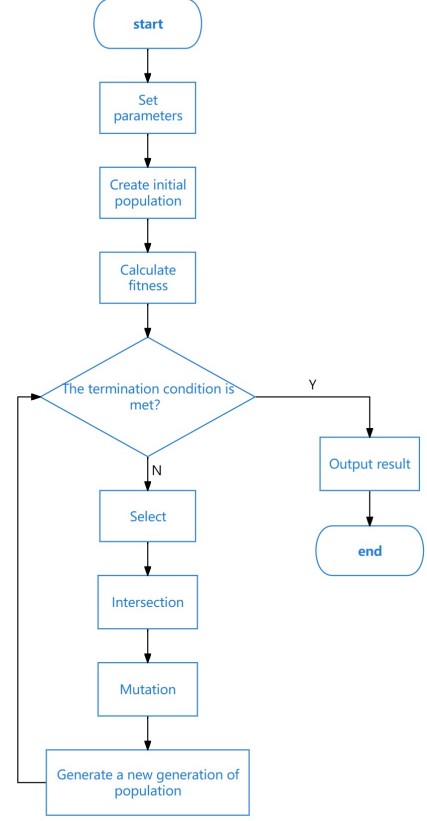

Fig. 2. Genetic Algorithm Flowchart

The basic principle of feature selection using GA is to find the optimal binary code using GA [7]. In the process of feature selection, all individuals in population are a string of binary numbers, the length of numbers is same as the number of features being selected, and each bit in code represents a feature. If the ith bit is 1, it means that the feature is selected; if the ith bit is 0, it means that the feature is not selected. After all coding are done, the inferiority or superiority of each individual is determined by calculating fitness function. After all the judgments are complete, the most adapted individuals, namely, the best individual in the population, are unconditionally copied into the next generation of the new population. Subsequently, genetic operators such as selection, intersection, and mutation are performed on the parent population to reproduce the next generation of the new population. If the set number of iterations is reached, the best gene string is returned and uses as basis for feature selection, and algorithm ends.

### C. XGBoost algorithms

The XG algorithm, developed by Tianqi Chen, is an ensemble method based on decision trees. Its primary objective is to

overcome computational limitations and achieve fast computation and excellent performance in engineering applications [15]. During model training, XG utilizes the classic boosting method and enhances the Gradient Boosting algorithm through second-order Taylor expansion, regularization term expansion, and coefficient operations. These enhancements establishes XG as a preferred tool for data scientists and engineers working with complex datasets [16].

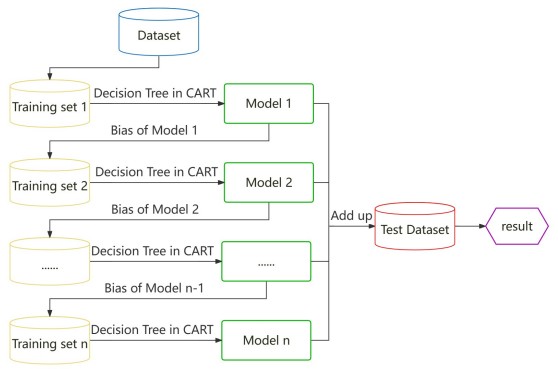

Fig. 3. XGBoost Algorithm schematic diagram

The most significant advantage of XG over other ensemble algorithms is computational speed. Tianqi Chen is experimentally demonstrated the superiority of the XG algorithm over other ensemble algorithms in terms of running time in the paper [15]. As base model for stacking models, XG is able to quickly train models and make predictions, reducing training time. So, in this paper, XG model is chosen as base model of the stacking model to minimize training time of the first layer of the model.

### D. RandomForest algorithms

RF is one of the most representative Bagging algorithms, and all its base evaluators are CART decision trees [17]. Before constructing the decision tree, the N samples in the dataset are randomly selected with replacement to obtain a new dataset of N samples for building the decision tree. When decision tree is split, m (m<M) of the M attributes that the sample is selected, and subsequently the optimal split point and split attributes are selected by information gain or Gini impurity. Generating a large number of decision trees according to this pattern gives the prototype of a random forest [18].

RF is a classical ensemble algorithm. Compared to other algorithms, there is a significant advantage in RF, which lies in its strong generalization ability [19]. It also doesn't require high data requirements and is not greatly affected by data problems such as feature loss and different quantities. As mentioned in, this makes it a reliable choice for various applications [20]. As a meta-model of the stacking model, RF can process different datasets more quickly and accurately, allowing it to output more accurate results. So, through theoretical support, RF is used to stack final output of the model.

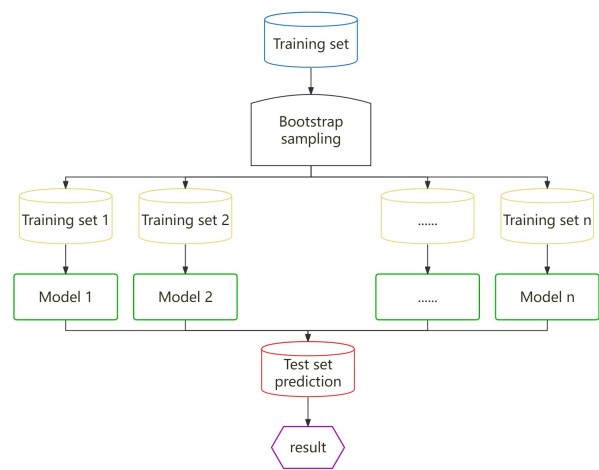

Fig. 4. RandomForest Algorithm schematic diagram

### III. GA-XG-RF MODEL STRUCTURE

This section describes overall structure of the model. The stacking model uses XG as base model of the stacking model and RF as meta-model of the stacking model. The advantage of using this structure is the ability to take full advantage of both models. Although XG is faster, it is more sensitive to data and is more suitable for optimizing training data through GA in the first layer. RF is asscociated with small data requirements and can produce stable results for different datasets, so it is more suitable for final output of the model. The overall structure diagram of the model is as follows:

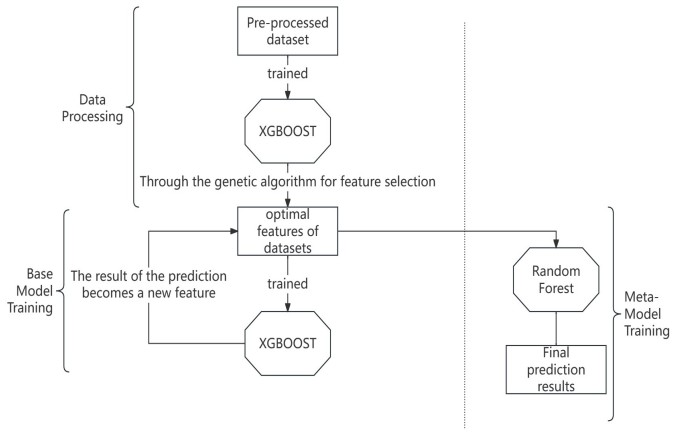

Fig. 5. The whole process structure diagram of the experiment

The whole model is divided into three parts: data processing, base model training and meta-model training. In this section, the principle and specific structure of each part will be introduced.

### A. Data processing

In the data processing section, the dataset is first trained using XG and GA with the aim of feature selection. In

the parameter setting process, fitness function of the genetic algorithm was chosen as the accuracy(ACC) value obtained after the XG training dataset. In order to avoid falling into the local optimal solution, the iterative method is used for optimization. In addition, the number of generations, crossover rate, mutation rate and other parameters should be set. In the evolutionary algorithm, when the fitness of the optimal individual in a continuous population of a certain generation does not change, the algorithm stops evolving and obtains the optimal feature combination in the calculation. This combination is used for next iteration of optimization, terminates after a certain number of iterations.

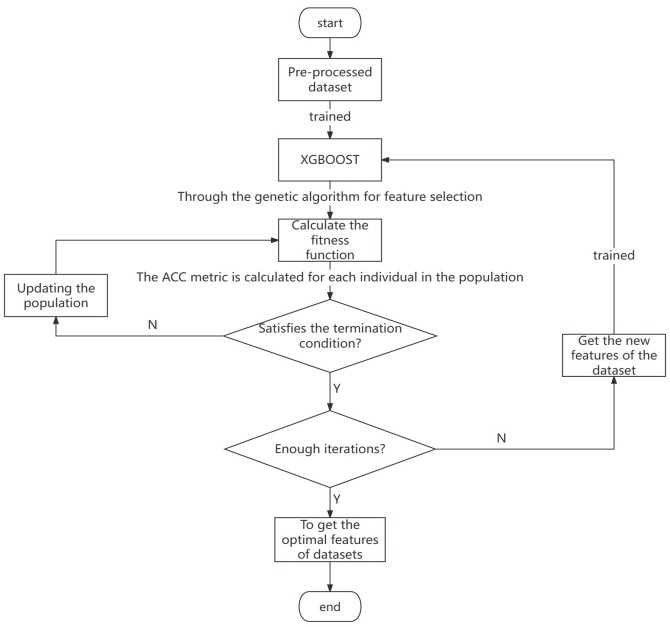

Fig. 6.  Supplementary llustration of data processing

## B. Base model training

When iteration is complete, a new set of features for XG model accuracy optimization can be obtained. Next, this new dataset is divided into k exclusive parts, and k identical XG models are initialized. Each is trained on k-1 subsets and tested using the remaining one. Note that each XG model is tested on a different subset.

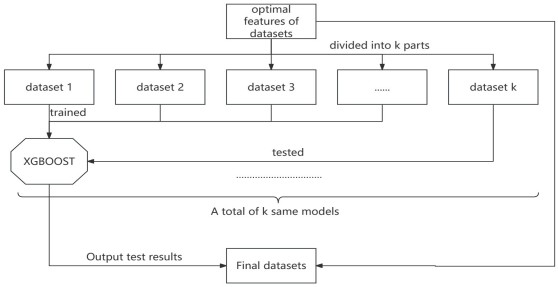

Fig. 7.  Supplementary llustration of base model training

## C. Meta-model training

The test results obtained using base model are merged into GA optimized dataset as a new feature. In this process, it is first necessary to combine the prediction results of k XG models, and the predicted results are added to the genetic algorithm optimized feature set as a new dataset. Finally, the new dataset is used to perform cross-validation on RF model and calculate average value of the performance index, and the GA-XG-RF stacking model is constructed.

## IV. EXPERIMENTAL RESULTS

In this experiment, three datasets from UCI and Kaggle are selected, namely, the heart disease dataset, ground occupation dataset, and wine quality classification dataset. These three datasets are all binary classification datasets. The performance indices and running time of each single model on the three datasets are first verified.

After some basic processing of the datasets, a single ensemble model is first used to train on three datasets, and four models, namely, AdaBoost, GradientBoostingMachine, XG, and RF, are chosen for this experiment to examine the running time and performance indices including ACC, PRE, REC, and F1 on different datasets.

As can be seen in the table I, the performance indices of RF are excellent when processing these datasets, while the performance indices of the other three models are close, But it's clear that XG's running time on all three datasets is more than 50% shorter than the other models, which the contribution it can make to the stacking model is predictable. Through this experiment, the theoretical advantages of RF and XG can be verified.

Next, in order to improve the accuracy and reduce the running time, GA is used to optimize the training process of XG. Using the optimal ACC value of XG as a fitness function, the corresponding feature set can be obtained. The XG model is trained with this set, and the running time and performance index of the model are obtained.

In the comparison experiment, the performance of three stacking models on each dataset is given in table II to table V. To facilitate tables, H dataset is for heart disease dataset, G dataset is for ground occupation dataset, and W dataset is for wine quality classification dataset.

As can be seen from table II, the AdaBoost-RF model and the non-GA-optimized XG-RF model performs well on the heart disease dataset, improving all performance measures by more than 5%. However, the data in Tables III to V prove that the performance of these two models on the other two datasets is not significantly different from that of the single model, and even lower than that of the single model in some indicators. In contrast, the GA-XG-RF stacking model shows excellent performance on all datasets. In three datasets, the GA-XG-RF stacking model obtained the best ACC, PRE, and F1 values among all experimental models. The GA-XG-RF stacking model also asscociated with at least 1% improvement in the part where there is no obvious difference between stacking model and single model, which indicates that there

TABLE I
PERFORMANCE INDICES AND RUNTIME OF A SINGLE ENSEMBLE MODEL ON DIFFERENT DATASETS

| Dataset | Number of features | Performance indices | RandomForest | AdaBoost | GradientBoostingMachine | XGBoost |
|---|---|---|---|---|---|---|
| Heart disease dataset | 17 | ACC | **0.763** | **0.763** | 0.733 | 0.741 |
| | | PRE | 0.743 | **0.758** | 0.723 | 0.736 |
| | | REC | **0.805** | 0.772 | 0.756 | 0.728 |
| | | F1 score | **0.773** | 0.765 | 0.739 | 0.755 |
| | | Running time | 126.83 | 329.35 | 1388.04 | **27.41** |
| Ground occupancy dataset | 5 | ACC | **0.978** | 0.943 | 0.952 | 0.945 |
| | | PRE | **0.938** | 0.859 | 0.890 | 0.903 |
| | | REC | **0.981** | 0.889 | 0.941 | 0.901 |
| | | F1 score | **0.957** | 0.857 | 0.911 | 0.892 |
| | | Running time | 16.22 | 13.60 | 44.44 | **1.42** |
| wine quality classification dataset | 11 | ACC | **0.727** | 0.697 | 0.696 | 0.696 |
| | | PRE | **0.760** | 0.751 | 0.731 | 0.724 |
| | | REC | **0.728** | 0.701 | 0.699 | 0.723 |
| | | F1 score | **0.730** | 0.707 | 0.704 | 0.713 |
| | | Running time | 9.08 | 10.66 | 42.56 | **4.64** |

TABLE II
PERFORMANCE OF THREE STACKING MODELS ON THE HEART DISEASE DATASET

| model | Used Features | ACC | PRE | REC | F1 score |
|---|---|---|---|---|---|
| Adaboost-RF | 17 | 0.818 | 0.802 | **0.846** | **0.823** |
| XG-RF | 17 | 0.813 | 0.796 | 0.845 | 0.819 |
| GA-XG-RF | 11 | **0.820** | **0.809** | 0.839 | **0.823** |

TABLE IV
PERFORMANCE OF THREE STACKING MODELS ON THE WINE QUALITY CLASSIFICATION DATASET

| model | Used Features | ACC | PRE | REC | F1 score |
|---|---|---|---|---|---|
| Adaboost-RF | 11 | 0.731 | 0.759 | 0.731 | 0.735 |
| XG-RF | 11 | 0.728 | 0.760 | **0.735** | 0.737 |
| GA-XG-RF | 7 | **0.741** | **0.774** | 0.734 | **0.746** |

TABLE III
PERFORMANCE OF THREE STACKING MODELS ON THE GROUND OCCUPANCY DATASET

| model | Used Features | ACC | PRE | REC | F1 score |
|---|---|---|---|---|---|
| Adaboost-RF | 5 | 0.971 | 0.918 | 0.982 | 0.946 |
| XG-RF | 5 | 0.976 | 0.928 | **0.986** | 0.954 |
| GA-XG-RF | 2 | **0.986** | **0.963** | 0.980 | **0.971** |

TABLE V
COMPARISON OF RUNNING TIME OF EACH STACKING MODELS ON DIFFERENT DATASETS

| model | H dataset | G dataset | W dataset |
|---|---|---|---|
| Adaboost-RF | 120.88 | 17.45 | 12.73 |
| XG-RF | 99.72 | 13.81 | 12.30 |
| GA-XG-RF | **83.56** | **12.24** | **10.32** |

is obvious robustness and superiority in test accuracy in the GA-XG-RF stacking model.

In addition, there is significant optimization in terms of running time in the GA-XG-RF stacking model. It can be seen from the table that the running time of the GA-XG-RF stacking model is the smallest among three stacking models. In addition, given the computational complexity of a stacking model compared to a single model, the running time is necessarily extended. In this case, the running time of the GA-XG-RF stacking model can also exceed single model of RF, Adaboost, and GradientBoostingMachine, which is enough to show the fast running time of the GA-XG-RF stacking model.

In summary, the GA-XG-RF stacking model can be optimized for both running time and computational accuracy.

Compared with single model, GA-XG-RF model can greatly improve the performance indices while taking into account the fast running time. Compared with other stacking models, GA-XG-RF model can improve performance indices while greatly optimizing running time and truly achieving comprehensive time and accuracy optimization.

## V. Conclusion

To optimize the stacking model in terms of running time and performance indicators, this paper proposes the GA-XG-RF stacking model, which uses XG and RF as the base model and meta-model, respectively, and pre-optimizes the XG training process using GA, and finally validates it using various datasets. The results demonstrate that the model can improve performance within a shorter running time. In the subsequent work, the improvement of RF is a focus, and the improvement of RF itself may further improve the strength of the model.

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
