# OpenReview forum: "Fast Classification Model Based on Genetic Algorithm and XGBoost-RandomForest Stacking Model"
_IEEE.org/ICIST/2024/Conference — IEEE ICIST 2024 Conference Submission_

### Official Review · Reviewer_uGzm · 2024-08-25
**minor repair**

**Rating:** 8
**Confidence:** 3

**Review:**

1. Authors should summarize the contributions in the Introduction Part introducing comparison with pervious researches.
2. The effectiveness of our proposed methods should be validated through comparisons.
3. The English should be improved.

---

### Official Review · Reviewer_ZtRW · 2024-08-29
**Review on Fast Classification Model Based on Genetic Algorithm and XGBoost-RandomForest Stacking Model**

**Rating:** 7
**Confidence:** 4

**Review:**

1. The authors should check the whole paper and avoid the grammar mistakes. For example, '' The main contribution of this paper are as follows''.
2. What are the meanings of PRE, REC, and F1?
3. The authors claim that in order to avoid falling into the local optimal solution, the iterative method is used for optimization. Can the local optimal solution be avoided? How to prove it?
4. The researches on other stacking models are missing in related work

---

### Official Review · Reviewer_Mgoj · 2024-09-01
**Comments to paper 2**

**Rating:** 7
**Confidence:** 3

**Review:**

In this paper, a hybrid ensemble learning model is introduced to improve computational accuracy, and genetic algorithm is used to select features in the training process to reduce running time. The authors need to address the following questions:
1. Comparative results are necessary to validate the claimed advantages of the proposed approach.
2. The simulation results, while intriguing, cannot be replicated without more information on
test set up, constants, etc.
3. Some typos and grammar errors should be modified in this paper. Please carefully double check
the manuscript.

---

### Decision · Program_Chairs · 2024-09-06

Accept (Oral)